# Drivers of Antibiotic Resistance Gene Abundance in an Urban River

**DOI:** 10.3390/antibiotics12081270

**Published:** 2023-08-01

**Authors:** Joseph C. Morina, Rima B. Franklin

**Affiliations:** Department of Biology, Virginia Commonwealth University, Richmond, VA 23284, USA; rbfranklin@vcu.edu

**Keywords:** antibiotics, ARGs, combined sewage overflow, CSO, wastewater, sewage, tetracycline, ß-lactams, quinolones, *tetO*, *tetW*, *bla_TEM_*, *qnrA*, *ampC*

## Abstract

In this study, we sought to profile the abundances and drivers of antibiotic resistance genes in an urban river impacted by combined sewage overflow (CSO) events. Water samples were collected weekly during the summer for two years; then, quantitative PCR was applied to determine the abundance of resistance genes associated with tetracycline, quinolones, and β-lactam antibiotics. In addition to sampling a CSO-impacted site near the city center, we also sampled a less urban site ~12 km upstream with no proximal sewage inputs. The tetracycline genes *tetO* and *tetW* were rarely found upstream, but were common at the CSO-impacted site, suggesting that the primary source was untreated sewage. In contrast, *ampC* was detected in all samples indicating a more consistent and diffuse source. The two other genes, *qnrA* and *bla_TEM_*, were present in only 40–50% of samples and showed more nuanced spatiotemporal patterns consistent with upstream agricultural inputs. The results of this study highlight the complex sources of ARGs in urban riverine ecosystems, and that interdisciplinary collaborations across diverse groups of stakeholders are necessary to combat the emerging threat of antibiotic resistance through anthropogenic pollution.

## 1. Introduction

Antibiotic resistance is an emerging global public health concern [1,2]. In 2019, 1.3 million deaths were directly attributable to antibiotic resistance, and an estimated total of 5 million deaths were associated with resistant bacteria [3]. Antibiotic resistance genes (ARGs), which confer resistance to specific antibiotics through a variety of mechanisms, occur naturally to mediate resource competition among microorganisms. However, the increased use of antibiotics as clinical agents has radically increased the evolution and spread of ARGs. Although most efforts to contain the antibiotic resistance problem are focused on biomedical settings, ARGs are rapidly emerging as environmental contaminants. There has been a particular focus on urban rivers as environmental reservoirs because they are a vital source of fresh drinking water and widely used for recreation. Sources of ARGs in these waterways include municipal wastewater systems [4], effluent from hospitals [5], pharmaceutical manufacturing [6,7], and upstream agricultural runoff [8].

Urban waterways and rivers are well-known hotspots for ARGs [9,10]. This is often attributed to fecal contamination, the introduction of antibiotics or metals that apply selective pressure to microbial communities, and horizontal gene transfer of the mobile genetic elements conferring antibiotic resistance [11]. Even in urban environments with extensive sanitation systems, the fact that wastewater treatment plants (WWTP) lack antibiotic-targeted treatment processes means that receiving waterbodies can be reservoirs of residual antibiotics and ARGs. In many older cities, the problem is compounded by the discharge of untreated wastewater from combined sewage–stormwater systems that overflow during heavy rain events. These combined sewage overflow (CSO) events appear to be a major source of antibiotics, resistance genes, and resistant organisms to urban rivers [12,13,14]. Since these rivers can act as conduits of antibiotics and ARGs to downstream aquatic and coastal systems, these wastewater inputs can have lasting repercussions across broad spatial scales.

A recent global meta-analysis found sanitation infrastructure to be one of the strongest drivers of elevated antimicrobial resistance, having a much greater effect than antibiotic consumption rates [15]. Although numerous studies have considered the impact of wastewater on the distribution and diversity of ARGs in urban rivers [4,16,17,18,19], the specific impact of combined sewage systems has received comparatively less attention [20,21]. Combined sewage systems are common in older cities around the world, especially in Europe and the United States. For example, the most recent estimates from the U.S. Environmental Protection Agency are that 40 million Americans live in municipalities with combined sewage–stormwater systems, which collectively discharge 850 billion gallons of contaminated wastewater each year [22]. Understanding the role of CSOs in the environmental dissemination of ARGs is becoming increasingly important as urbanization and climate change can lead to even more frequent and intense overflow events [23,24].

The purpose of this study was to document the prevalence of ARGs and their relationship with water quality parameters in an urbanized segment of the James River (Virginia, USA) that is frequently impacted by CSO events. We compared a site upstream of major urbanization (“HUG”) to one located near a large CSO outfall at the city center (“CSO”). The abundances of five ARGs were determined using quantitative polymerase chain reaction (qPCR) and compared with changes in water quality and hydrological conditions to identify potential sources of ARG within the watershed. We focused on summer when recreational contact, and thus the potential for human health impacts, is highest.

## 2. Results and Discussion

Fecal contamination is one of the main sources of ARGs in urban aquatic environments [25], in part due to increased population density and high levels of impervious surfaces [26,27]. Combined sewage overflow systems in urban settings are known to be a major source of ARGs and resistant bacteria to receiving waterbodies [12,14,28]. Our results suggest that while CSO events are likely a large source of ARGs in the river, non-point sources are also important sources of ARGs in the river. Overall, ARGs were more abundant at the downstream CSO site than the upstream site (HUG) (Figure 1).

Tetracycline-resistance genes showed the most striking differences across sites. The *tetO* gene was never detected at the upstream site and the *tetW* gene was only detected in ~2% of samples, whereas these genes were detected in 35% (*tetO*) to 48% (*tetW*) of samples from the CSO site. Mann–Whitney tests confirmed elevated abundances of both genes at the CSO site (*tetO*: U = 243, *p* = 0.001; *tetW*: U = 424, *p* < 0.0001). Correlation analysis showed abundances of both tetracycline genes were most strongly related to *E. coli* abundance (Table 1), suggesting fecal contamination is a major source of these ARGs in the river. The increase in abundance of *tetO* at the CSO site also correlated with an increase in TN (ρ = 0.60) and TP (ρ = 0.67). Since increased concentrations of N and P have been well documented in association with CSO events [29,30], this correlation provides further support that *tetO* derived from fecal contamination due to sewage overflow. In contrast, the increase in abundance of *tetW* at the CSO site was correlated with precipitation, suggesting non-point sources of fecal contamination via runoff or leaky sewerage [31].

Tetracycline resistance genes (*tet*) are some of the most commonly detected and diverse ARGs in aquatic habitats, with over 20 types identified thus far [32]. These genes are frequently found in animal and human feces [33], are highly abundant in sewage [34,35,36], and their presence in surface waters has been linked to fecal contamination at the national level in the United States [37]. The abundance of *tet* genes has also been found to be elevated downstream of WWTPs [38]. Our findings are consistent with those prior studies: both of the *tet* genes we considered were found in high abundance at the CSO site when the river was contaminated with fecal material due to either sewage overflow events (*tetO*) or runoff (*tetW)*). Matsui and Miki [13] also studied multiple *tet* genes in an urban river and, like us, found some to be linked with rainfall and non-point discharge (*tetM*), whereas others were not (*tetA* and *tetB*). Their study also demonstrated that increasing the storage of CSO systems can decrease the spread of *tet* genes. Our data suggest that similar actions would decrease the spread of *tetO* but may not impact *tetW*. This highlights the need for studies such as ours that seek to differentiate point and non-point sources of ARG pollution within the urban landscape, as each may ultimately require different management approaches.

In addition to being common in sewage, *tet* genes are a major component of soil resistomes [39,40]. Given this, it was somewhat surprising to find both tetracycline genes to be essentially absent at the upstream site. The widespread distribution of *tet* genes in soil is likely due, at least in part, to the fact that many code for efflux pumps that can serve diverse functions in addition to conferring antibiotic resistance [32]. However, *tetO* and *tetW* are unusual in that they code for ribosomal protection proteins; thus, their presence/absence is likely more reflective of tetracycline exposure compared to other *tet* genes. Our failure to find *tetW* and *tetO* at the upstream site suggests there is not a significant proximal upstream source of human waste or any nearby agricultural operations that utilize tetracycline. Although tetracyclines are one of the primarily antibiotics groups used for agricultural purposes [41], most of the nearby farms are equine, and the use of tetracycline antibiotics to treat horses is discouraged because doing so can cause severe and even fatal diarrhea [42,43].

The *bla_TEM_* gene, which encodes for the TEM-1 β-lactamase, confers resistance to β-lactam antibiotics such as the penicillins and early cephalosporins. This gene was found in ~55% of all samples and showed similar abundances across the two sites (U = 766, *p* = 0.90). Of the five ARGs we surveyed, only *bla_TEM_* was abundant at the upstream site (Figure 1). Approximately ~30% of HUG samples had > 6000 copies mL^−1^ of *bla_TEM_* (15% had > 10,000 copies mL^−1^), whereas the abundance of the four other ARGs never exceeded 4500 copies mL^−1^ at the upstream site. The high overall abundance and broad distribution of the *bla_TEM_* gene were not surprising, as this gene appears to be one of the most widespread ARGs across multiple environmental compartments [44,45]. This is further supported by a recent survey of 2000 streams and rivers across the USA, which found *bla_TEM_* to be ubiquitous across all ecoregions of the country, despite different land-use regimes and environmental conditions [37]. This widespread distribution of *bla_TEM_* is likely due to the fact ß-lactams were the first major class of antibiotics to be discovered and account for approximately two-thirds, by weight, of all antibiotics prescribed to humans [46]. Like *tetW*, we found *bla_TEM_* abundances to be correlated with *E. coli* (ρ = 0.48) and prior-day precipitation (ρ = 0.46). This suggests upstream or catchment sources of *bla_TEM_* in the river, likely linked to fecal contamination being introduced into the river during rain events via runoff. Similar research examining the drivers of ARGs in a riverine system found a strong link between ARG abundances and precipitation, suggesting the catchment area is a source of ARGs in the river [16]. The remaining two genes, *ampC* and *qnrA,* did not show a direct link with fecal contamination or sewage overflow indicators, and were similarly distributed across the two sites (Figure 1). The *ampC* gene, which also confers resistance to multiple members of the ß-lactams class of antibiotics, was the only ARG that we detected in every sample. Although the Mann–Whitney test indicated elevated abundance of *ampC* at the CSO site (U = 656, *p* = 0.01), the difference was relatively small (median and SE-median for HUG: 919 ± 211; for CSO: 1412 ± 421), especially compared to the abundance of other ARGs. The abundance of *ampC* was positively correlated with overall bacterial abundance, but did not appear to be related to any of the water quality or hydrologic parameters we measured. Coertze and Bezuidenhout [47,48] also found it challenging to correlate environmental factors or land use as a predictor of *ampC* abundance, and speculated that there must be a pervasive source or reservoir of this ARG in the rivers they studied in South Africa. Something similar could be occurring in our system, though the positive correlation of *ampC* and bacterial abundance suggests that *ampC* is simply widespread in the James River bacterial community. Our results indicate that *ampC* is a highly dispersed gene, consistent with reports of broad global distribution of the gene across terrestrial [39,49] and aquatic [50] ecosystems, including relatively pristine or unimpacted sites [51,52].

Of the five ARGs we considered, *qnrA*, which confers resistance to fluoroquinolones, was the least common overall. The *qnrA* gene was only detected in ~40% of samples, and abundance never exceeded 3000 copies mL^−1^ (Figure 1). No differences were detected across sites (U = 697, *p* = 0.48), and *qnrA* abundances were not correlated with either *E. coli* abundance or TN and TP concentrations (Table 1). Together, these results suggest that sewage was not a significant reservoir of *qnrA*, despite the fact that fluoroquinolones are frequently prescribed to humans. This finding may be due to the fact that we sampled exclusively during summer. Seasonal variation in fluoroquinolone concentration has repeatedly been found in freshwater rivers [53], WWTP influent [54], and sludge from WWTPs [55], with levels being consistently higher in winter. This indicates changing prescription and usage levels of fluoroquinolones in human medicine [56], and suggests that our *qnrA* results could be very different if we sampled during winter months.

Correlation analysis did reveal a positive relationship between *qnrA* abundance and rainfall the day prior to sampling (ρ = 0.43), which suggests that precipitation mobilizes an upstream or catchment-level source of this resistance gene to the James River. The watershed upstream of the HUG site is approximately 10% pasture land [57], and there are multiple equestrian centers, with the closest being ~10 km upstream, making this a plausible source of *qnrA* genes or fluoroquinolones for the river. Antimicrobial resistance is prevalent in bacteria from horses [58], and *qnr* genes have previously been detected in the feces of horses and in environmental samples from equine clinics and horseback riding centers [59,60,61]. Our hypothesis that equestrian facilities are the most likely agricultural source of *qnrA* is also supported by the fact that fluoroquinolones are widely utilized in animal husbandry and in equestrian activities/rearing [59,62], but rarely used in food-producing animals (~0.1% of U.S. domestic antibiotic sales during the study period [63]).

Interestingly, of the ARGs that correlated with precipitation, the *qnrA* gene was the only one that was not also linked with *E. coli* abundance. This could suggest that the fluoroquinolone-resistant bacteria survive longer in the riverine system than the fecal indicator bacteria, or that the plasmid containing this ARG is especially persistent. This persistence could be due to the selective pressure of fluoroquinolones in the river water. For example, Stanton et al. [64] documented that ciprofloxacin can select for resistance persistence even at the relatively low concentrations reported in aquatic environments. This effect may be compounded by the fact that fluoroquinolones can remain stable in river water for at least two weeks [65]. Future work is needed to determine fluoroquinolone concentrations in the James River, but these prior studies suggest that low antibiotic concentrations may be decreasing the rate at which resistance genes disappear from the environment. These findings are particularly concerning, because the increased persistence of resistant bacteria or resistance genes is likely to associated with higher human exposure risk and greater impact on environmental microbiomes.

## 3. Conclusions

Taken together, our efforts show that over a relatively small spatial scale (~12 km), abundances of ARGs can differ by 3–5 orders of magnitude. Our multi-year dataset encompassing a variety of weather conditions shows diverse sources, including urban runoff, CSO events, and upstream land use, contribute to ARG abundances in urban rivers. Furthermore, our results show spatiotemporal considerations should be incorporated into investigations aiming to elucidate sources of ARGs to urban rivers or waterways. Future work that considers additional genes such as *intI1*, which has been used as a proxy indicator of anthropogenic pollution [37], could help further distinguish among the potential ARG sources and would provide valuable information regarding the potential for ARG spread by horizontal gene transfer.

While efforts to improve wastewater treatment technology and infrastructure will certainly reduce ARG abundance and antibiotic pollution in urban aquatic ecosystems, our results suggest that levels of select ARGs could remain elevated due to non-point sources within the urban area and from the surrounding landscape. Such a complex and multifaceted public health concern requires collaboration between environmental scientists, public health officials, wastewater management authorities, agricultural stakeholders, and policy makers in order to address the emerging threat of antibiotic resistance in aquatic environments.

## 4. Materials and Methods

### 4.1. Site Description

Richmond, Virginia (USA) is a moderately sized city with approximately 226,000 residents and a population density of ~10,000 people per km^2^ (United States Census Bureau, 2016). It relies on a combined sewage–stormwater system, which overflows and discharges untreated wastewater into the James River during heavy rainfall or snow melt. Richmond’s combined sewer system is the largest in the state of Virginia, and services approximately one-third of the city (~50 km^2^).

This study compared two sites along the James River as it flows through Richmond (Figure 2). The first was located within the Huguenot Flatwater area of James River Park (HUG; 37.560471, −77.545801), approximately 12 km upstream of the city center. This site was selected to assess water quality before significant urbanization and prior to any known CSO outfalls. The watershed upstream of HUG is forested and agricultural land, but also includes the cities of Lynchburg (~225 km upstream with 80,000 residents in 2016) and Charlottesville (~150 km upstream with 47,000 residents) and several smaller municipalities.

The second site was located near the city center, adjacent to the city’s largest CSO outflow (37.529486, −77.429382), which is referred to as CSO-006 by the Richmond Department of Public Utilities. Long-term monitoring of *Escherichia coli* (*E. coli*) abundance indicates consistently higher levels of fecal contamination at this site compared to HUG. During the two-year sampling period of this study, *E. coli* concentrations exceeded 200 CFU 100 mL^−1^ in 50% of CSO samples versus <25% of HUG samples (Figure 1). The maximum *E. coli* abundance observed at the CSO site was approximately four-times greater than at HUG (32,200 versus 800 CFU 100 mL^−1^).

### 4.2. Water Sampling

Approximately every week during the summers (May 1st through October 15th) of 2015 and 2016, surface water samples (*n* = 44) were collected from each site using a bucket thrown from shore. Water was transferred into sterile plastic 1 L bottles and transported back to lab on ice within two hours. Upon return to the lab, samples were immediately processed to determine *E. coli* abundance following the EPA Method #1603 [66]. Plate counts were performed using modified mTEC agar (BD Difco, Sparks, MD, USA), and the results are reported as colony-forming units (CFU) 100 mL^−1^ of river water. In addition, aliquots (300 mL) of water were filtered using 0.2-µm pore-size polycarbonate membranes (Millipore, Molsheim, France) to isolate the microbial community for genetic analysis. Filters were stored at −20 °C until DNA extraction could be performed.

### 4.3. Environmental Data

Total nitrogen (TN) and total phosphorus (TP) concentrations (Figure 3) were determined as part of a long-term monitoring program using previously published analytical methods [67]. Discharge data (Figure 4) were obtained from the United States Geological Survey (USGS) gaging station at site 02,037,500 (37.563055, −77.547222), which is adjacent the HUG sampling site. Precipitation data (Figure 5) were obtained from the National Climatic Data Center using the Richmond International Airport site, “KRIC”.

### 4.4. DNA Extractions

Extractions were performed using the PowerWater DNA Isolation Kit (MoBio Laboratories, Carlsbad, CA, USA). The following modifications were made to the manufacturer’s protocol to increase extraction efficiency. First, each filter was torn into small pieces using sterile forceps prior to being inserted into the PowerWater Bead tube. Next, to minimize DNA shearing, all vortex speeds were reduced to the lowest possible speed that still allowed for mixing. The incubation step for the removal of non-DNA organic and inorganic matter was extended to 10 min. Lastly, the elution step, normally one 100 µL elution with no incubation, was divided into two 50 µL elutions with an additional 5-min incubation at room temperature before each centrifugation. Successful extraction was determined using agarose gel electrophoresis (1.5%) and ethidium bromide staining, and DNA concentration was measured using Quant-iT PicoGreen dsDNA Assay Kit (Invitrogen, Carlsbad, CA, USA).

### 4.5. Quantitative Polymerase Chain Reaction (qPCR)

Total bacterial abundance was determined by quantifying the *16S rRNA* gene with the primers Eub338/Eub518 following previously reported reaction conditions [68]. We quantified the abundance of five ARGs: *tetO* and *tetW* for resistance to tetracyclines, *bla_TEM_* and *ampC* for β-lactam antibiotics, and *qnrA* for quinolones. These ARGs confer resistance to some the top prescribed antibiotics in the United States [69], and bacteria with resistance to the respective antibiotics have previously been detected at the two sites [12]. Each qPCR run included an appropriate standard curve that covered at least eight orders of magnitude, with the lowest starting point being 88 (*tetO*), 32 (*tetW*), 44 (*bla_TEM_*), 51 (*ampC*), and 156 (*qnrA*) gene copies per reaction. For the *16S rRNA* gene assay, the curve was constructed using genomic DNA extracted from *E. coli* Strain NCTC 9001 (ATCC, Manassas, Virginia, USA). For the ARGs, standard curves were constructed using plasmid DNA extracted using the Zyppy Plasmid Miniprep Kit (Zymo Research Corp, Irvine, California, USA). Sources were: SpyTag-β-Lactamase-Spycatcher (pET28a) (Addgene, Cambridge, Massachusetts, USA) for *bla_TEM_*; pTrcHis + *qnrA* in *E. coli* J53 (Thermo Fisher Scientific, Waltham, Massachusetts, USA), and pCR^®^2.1-TOPO + either *tetW*, *tetO*, or *ampC* in DH5α *E. coli* (Invitrogen, Carlsbad, CA, USA). Each qPCR run also included two types of negative control: (i) a “negative template control”, which replaced sample DNA with nuclease-free water; and (ii) a “gene-free control”, which replaced sample DNA with genomic DNA from *Methanococcus voltae* (DSM #1537, DSMZ, Braunschweig, Germany).

Each qPCR assay was performed in triplicate (*qnrA* performed in quadruplicate) using a CFX384 Real-Time System (Bio-Rad, Hercules, CA, USA) and Bio-Rad SsoAdvanced Universal SYBR Green Supermix (Bio-Rad, Hercules, CA, USA). Reaction mixtures (15 µL) also contained 5 ng of template DNA and the appropriate concentration of forward and reverse primer (Integrated DNA Technologies, Coralville, IA, USA). Reaction conditions are presented in Table 2. A melt curve and agarose gel electrophoresis were conducted to verify the specificity of the amplified products. Amplification efficiencies ranged from 93–103% and all r^2^ > 0.98. Gene abundance data are reported as copies mL^−1^ of river water.

### 4.6. Data Analyses

All statistical tests and visualizations were performed using R (version 4.2.2) [75]. Non-parametric approaches were employed because ARG data were not normally distributed (Shapiro–Wilks tests, all *p* > 0.05). Alpha of 0.05 was used for all tests except correlations, which used a more conservative value of 0.01 to account for multiple comparisons.

Histograms were used to visualize the ARG data by plotting the fraction of samples (%) in each abundance interval (Figure 1). For simplicity and to facilitate statistical comparisons, data below the detection limit are interpreted as zeros. Mann–Whitney tests were performed to determine whether ARG abundances differed between the upstream site (HUG) and at the city center (CSO). Spearman correlation was then used to identify potential drivers of increased ARG abundance at the CSO site. We specifically considered covariation with bacterial abundance (*16s rRNA* copies mL^−1^), level of fecal contamination (*E. coli* CFU 100 mL^−1^), river discharge (m^3^ s^−1^), precipitation (cm), and nutrient concentrations. For precipitation, we evaluated two timeframes: “sampling-day precipitation” (cumulative rainfall from midnight on sampling day until sampling time (~12 h)) and “prior-day precipitation” (rainfall that fell between 12 and 36 h prior to sampling). For nutrients, we attributed increased concentrations of TN and TP at the CSO site to sewage overflow, and used the difference (Δ) as an indicator of the magnitude of the overflow event.

## Figures and Tables

**Figure 1 antibiotics-12-01270-f001:**
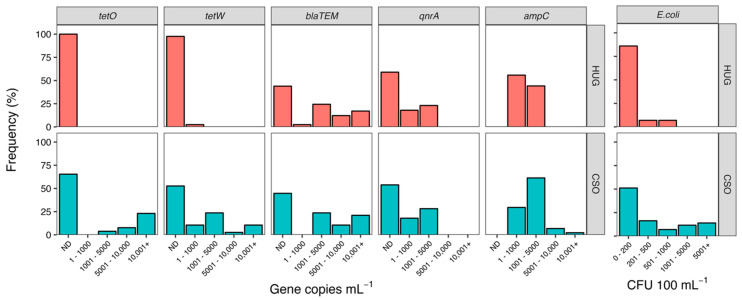
Abundance frequency histograms for ARG (**left**) and *E. coli* (**right**). Data are presented as the relative frequency (% of samples) in each abundance interval during the two-year study period. The top row (pink bars) shows data from the upstream site (HUG) and the bottom row (blue bars) shows the downstream site (CSO) near the city center. The not detected (ND) category represents samples that did not produce amplicons.

**Figure 2 antibiotics-12-01270-f002:**
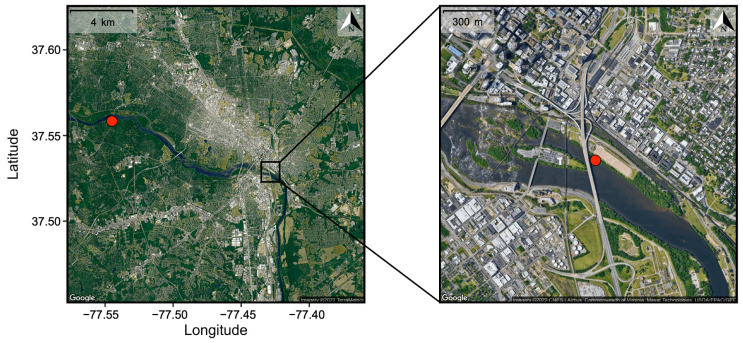
Map of the James River along Richmond, Virginia (USA). Red circles mark the upstream (HUG, **left panel**) and downstream (CSO, **right panel**) sampling sites, which are separated by ~12 km. The black insert box, expanded on the right, shows the extensive urbanization surrounding the downstream CSO site.

**Figure 3 antibiotics-12-01270-f003:**
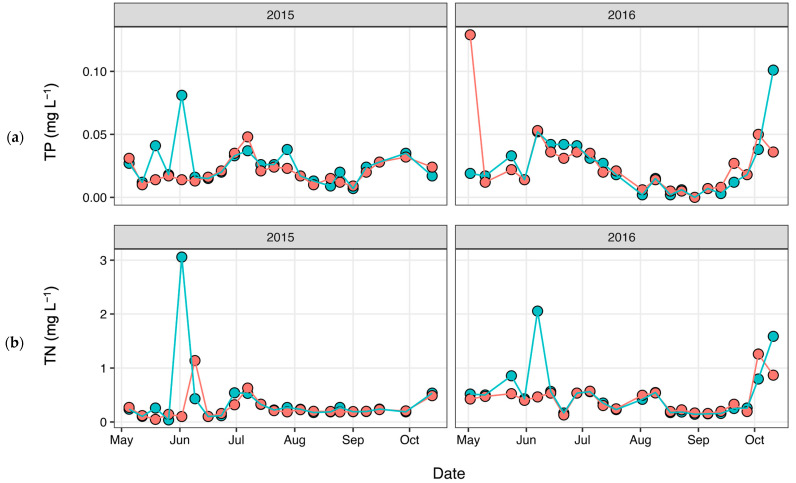
(**a**) Total phosphorus (TP) and (**b**) total nitrogen (TN) concentrations measured during each sampling event, with pink corresponding to HUG and blue corresponding to CSO.

**Figure 4 antibiotics-12-01270-f004:**
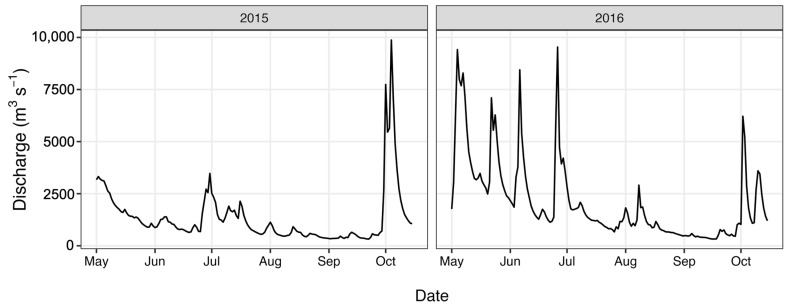
River discharge values during the study period.

**Figure 5 antibiotics-12-01270-f005:**
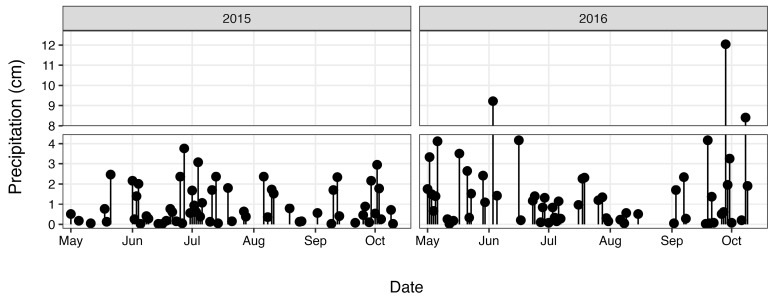
Precipitation during the study period.

**Table 1 antibiotics-12-01270-t001:** Spearman’s correlation coefficients (ρ) for ARG abundances and environmental parameters at the CSO site. Bold values represent statistically significant coefficients (α = 0.01), and asterisks reflect degree of significance (* 0.001 < *p* ≤ 0.01, ** 0.001 < *p*).

	*tetO*	*tetW*	*bla_TEM_*	*qnrA*	*ampC*
**Fecal Contamination Indicator**					
*E. coli* abundance	**0.85** **	**0.58** **	**0.48** **	0.15	0.23
**Sewage Overflow Indicators**					
Δ TN	**0.60** *	0.11	−0.11	−0.11	−0.28
Δ TP	**0.67** **	0.16	0.17	0.20	0.04
**Discharge and Precipitation**					
Discharge	−0.07	−0.10	0.19	0.09	−0.16
Sampling-day precipitation	0.37	**0.46** *	0.38	0.29	0.05
Prior day-precipitation	0.41	**0.49** *	**0.46** *	**0.43** *	0.04
Bacterial Abundance	−0.10	0.10	0.18	−0.12	**0.58** **

**Table 2 antibiotics-12-01270-t002:** Primers and reaction conditions for qPCR assays.

Antibiotic Class	Gene	Primer Information			Thermal Conditions (°C)
		Names	Source	µM	
Tetracyclines	*tetW*	tetW-F; tetW-R	[70]	0.10	95° for 4 min, 40 cycles of 95° for 30 s, 57.4° for 15 s, 72° for 15 s
	*tetO*	tetO-FW; tetO-RW	[71]	0.20	94° for 5 min, 45 cycles of 94° for 30 s, 60° for 30 s, 72° for 30 s
β-lactams	*bla_TEM_*	blaTEM-FX; blaTEM-RX	[72]	0.30	95° for 5 min, 40 cycles of 95° for 15 s, 61° for 30 s, 72° for 30 s
	*ampC*	ampC-F; ampC-R	[73]	0.20	94° for 3 min, 40 cycles of 94° for 20 s, 58° for 20 s, 72° for 45 s
Quinolones	*qnrA*	qnrAf-RT; qnrAr-RT	[74]	0.10	95° for 3 min, 45 cycles of 95° for 15 s, 59.9° for 20 s

## Data Availability

Raw data are available on request from the corresponding author.

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
