# Peer review of "Drivers of Antibiotic Resistance Gene Abundance in an Urban River"

_antibiotics, 2023, doi:10.3390/antibiotics12081270_

Round 1

Reviewer 1 Report

The threat of antimicrobial resistance is one of the greatest threat to public health and I acknowledge it is important to survey environmental reservoirs of ARGs. However, studies such as this has been conducted in multiple geographical locations all across the world. It is important to compare concentrations of such ARGs with studies conducted in low and middle income countries (where AMR consequences is severe compared to in Virginia) so that future review articles can come up with a guideline to decide what concentration levels of certain ARGs is acceptable or what concentration says the threat of AMR is too high. Therefore I have few comments and want the authors to carefully address them.

Line 12-13. "Sources of ARGs......runoff". Please cite relevant articles. For example Thakali et al., 2020 (https://doi.org/10.3390/w13192733) have clearly demonstrated the sources such as hospital wastewater and wwtp in their study. Similarly rivers impacted by effluents of pharmaceutical companies have also been demonstrated to harbor high concentration of ARGs.

Line25-28. Please elaborate how poor water quality and sanitation infrastructures enhance Antibiotic resistance.Maybe a line or two to explain why.

Line 44. Abundance of five ARGs.....There are plenty of ARGs responsible for antibiotic resistance to different classes of antibiotic resistance. Monitoring certain ARGs can be biased. A framework to monitor ARGs in aquatic environments has been proposed by Keenum et al., 2022 (https://www.tandfonline.com/doi/full/10.1080/10643389.2021.2024739). The ARGs mentioned in this framework paper and this study varies. Therefore, I would like the authors to please add the reasons behind selecting those ARGs in this surveillance.

Fig.1. Please minimize the use of short forms. for example HUG is very confusing since it was described only in the materials and methods section but results and discussion appears first on the manuscript. If it's one word I don't recommend use of short forms or clearly mention it wherever it appears first in the manuscript.

Figure1. 0 concentration or below the limit of detection?

Line 88. It is not surprising that tetracycline resistant genes are less abundant in upstream sites far away. For example Thakali et al.,  2020 https://doi.org/10.3390/w12020450 have reported similar findings where tet or other ARGs were significantly less in upstream compared to midstream and downstream sites.

Results.The authors also quantified 16SRNA. However, it seems that data was not used in this study. In events such as sewer outflow due to rainfall or storm events wastewater or river water can be severely diluted. Normalization with 16S rRNA would have removed this bias and provided best comparison between CSO event and upstream. So I would recommend further analysis with normalization and report another figure with normalization with 16S rRNA. if normalization still follows same trend or makes no difference please clearly mention in results section as this could be significant.

Methods. Please report the limit of detection of each ARG assay. I do not think it should be 0 but should rather be below the limit of detection. This will give an idea for non-detects.

Figure 2. Horse farms are mentioned in the abstract highlighting the importance of author's hypothesis that horse farms were possible cause of ARGs. Therefore, please include horse farms in the maps to give readers a clear idea how the authors hypothesis could be true since microbial source tracking for horses have not been conducted to confirm  horse farm influence.

Line 53-54 Our results.....also contribute.  Please rephrase the line.

Author Response

Line 12-13. "Sources of ARGs......runoff". Please cite relevant articles. For example Thakali et al., 2020 (https://doi.org/10.3390/w13192733) have clearly demonstrated the sources such as hospital wastewater and wwtp in their study. Similarly rivers impacted by effluents of pharmaceutical companies have also been demonstrated to harbor high concentration of ARGs.

Thank you for providing the reference. We have added Thakali et al. 2021 and four other relevant citations to support our claims in lines 12-13.

Line25-28. Please elaborate how poor water quality and sanitation infrastructures enhance Antibiotic resistance. Maybe a line or two to explain why.

Our primary focus is infrastructure (CSOs), so we have removed this reference to water quality. The potential impact of infrastructure on antibiotic resistance and bacteriological water quality is explained in lines 14-21 of the revised manuscript.

Line 44. Abundance of five ARGs.....There are plenty of ARGs responsible for antibiotic resistance to different classes of antibiotic resistance. Monitoring certain ARGs can be biased. A framework to monitor ARGs in aquatic environments has been proposed by Keenum et al., 2022 (https://www.tandfonline.com/doi/full/10.1080/10643389.2021.2024739). The ARGs mentioned in this framework paper and this study varies. Therefore, I would like the authors to please add the reasons behind selecting those ARGs in this surveillance.

As requested, we have added the reasoning behind our primer selection (see lines 286-288 of the revised manuscript). Briefly, the choice was due to both USA antibiotic prescription patterns and previous work, which showed resistance to the selected antibiotics at our study sites. Several of the ARGs we used are part of the U.S. EPA National Rivers and Streams Assessment survey (Keely et al. 2019; https://doi.org/10.1021/acs.est.2c00813), which permits comparison of our results with a national survey of ~2000 U.S. river and stream sites.

Fig.1. Please minimize the use of short forms. for example HUG is very confusing since it was described only in the materials and methods section but results and discussion appears first on the manuscript. If it's one word I don't recommend use of short forms or clearly mention it wherever it appears first in the manuscript.

Based on your advice, we added a short sentence to the introduction (lines 45-47) to orient the reader to the two sites and define the site codes (HUG and CSO). We have been careful not to use any other short forms or variations of site names. Hopefully, this makes the manuscript easier to follow.

Line 88. It is not surprising that tetracycline resistant genes are less abundant in upstream sites far away. For example Thakali et al.,  2020 https://doi.org/10.3390/w12020450 have reported similar findings where tet or other ARGs were significantly less in upstream compared to midstream and downstream sites.

Thank you for providing this source; we have included Thakali et al. 2020 in lines 90-91.

Results. The authors also quantified 16SRNA. However, it seems that data was not used in this study. In events such as sewer outflow due to rainfall or storm events wastewater or river water can be severely diluted. Normalization with 16S rRNA would have removed this bias and provided best comparison between CSO event and upstream. So I would recommend further analysis with normalization and report another figure with normalization with 16S rRNA. if normalization still follows same trend or makes no difference please clearly mention in results section as this could be significant.

We did indeed use the 16s rRNA data in our correlation analysis (Table 1), similar to the work presented in Thakali et al. 2020 (doi.org/10.3390/w12020450). If we were to standardize our genes to 16s rRNA abundance, we would not be able to use this as a correlator in our analysis. As our correlation analysis was focused only on the downstream site, the cross-site bias mentioned by the reviewer is not a concern.

In addition, we chose not to standardize based on 16s rRNA because we wanted our ARG values to be based on physical exposure potential (i.e., water volume), comparable to other measures used in public health monitoring. For instance, E. coli and fecal coliforms are reported as CFUs per volume of water, so we presented the ARG abundances similarly, using gene copies per mL.

Methods. Please report the limit of detection of each ARG assay. I do not think it should be 0 but should rather be below the limit of detection. This will give an idea for non-detects.

The requested information has been added in lines 289-290 and 320-321.

Figure 2. Horse farms are mentioned in the abstract highlighting the importance of author's hypothesis that horse farms were possible cause of ARGs. Therefore, please include horse farms in the maps to give readers a clear idea how the authors hypothesis could be true since microbial source tracking for horses have not been conducted to confirm  horse farm influence.

The reviewer makes an excellent point here, namely that by including horse farms in the abstract, we set the reader up to think this is a major takeaway from our paper. In fact, it is an untested hypothesized source and a relatively minor point. We opted to remove the reference to equine activities in the abstract.

Line 53-54 Our results.....also contribute.  Please rephrase the line.

This line was rephrased (lines 57-59 in this version of the manuscript).

Reviewer 2 Report

The manuscript deals with current issues related to the phenomenon of transfer of antibiotic resistance genes between different environmental reservoirs. The authors document the presence of selected antibiotic resistance genes (ARGs) in surface water samples collected in two summer seasons from the river at the point of sewage overflow and above it.

The work does not discuss the relationship between all the parameters studied, such as the concentration of total nitrogen and phosphorus. The results presented in Figure 1 are not very clear. Try to present them in a more interesting way. The cited literature should also be supplemented with the latest reports. There are many papers documenting sewage as a source of ARGs. The authors did not investigate the presence of genes encoding integrase. Such research could show that wastewater is a potential site for acquiring antibiotic resistance by microorganisms. Link this in a discussion with other authors.

Author Response

The work does not discuss the relationship between all the parameters studied, such as the concentration of total nitrogen and phosphorus.

Further discussion of TP and TN, and relevant citations, have been added to the manuscript (lines 77-79).

The results presented in Figure 1 are not very clear. Try to present them in a more interesting way.

We used multiple approaches to visualize this dataset, and concluded that the current version is the most appropriate for our objectives. However, we appreciate that the reviewer did not find the presentation to be clear, so we added clarifying language to the Figure 1 legend (lines 65-66) and in the methods section (lines 319-321). We also edited the y axis title to be more informative.

The cited literature should also be supplemented with the latest reports. There are many papers documenting sewage as a source of ARGs.

We have added a total of 12 new sources to the manuscript, including recent sources throughout to address this concern (lines 12-14, 35, 90-91, 94-101).

The authors did not investigate the presence of genes encoding integrase. Such research could show that wastewater is a potential site for acquiring antibiotic resistance by microorganisms. Link this in a discussion with other authors.

At the time the research was conducted, the use of intl1 as a proxy for anthropogenic pollution had just been proposed (Gillings et al., 2015). We agree this would be an exciting addition, but samples are no longer available for analysis. Following the reviewer’s recommendation, we have added a discussion of the integrase to lines 197-201 of the revised manuscript.

Reviewer 3 Report

Authors has highlighted the complex sources of ARGs impacting urban rivers. The findings indicate that untreated sewage from CSO events contributes significantly to tetracycline resistance gene abundance, while other genes have more diverse sources, including upstream agricultural inputs.

Minor comments

In introduction section, Emphasize the significance of untreated sewage as a primary source of tetracycline resistance genes (tetO and tetW) in the urban river impacted by combined sewage overflow (CSO) events.

Also Highlight the need for improved wastewater treatment to mitigate the introduction of antibiotic resistance genes (ARGs) into the aquatic environment

Discuss the potential health implications of high abundances of tetracycline resistance genes in the CSO-impacted site. Explore the possibility of gene transfer and the spread of antibiotic resistance among bacteria in the environment, posing risks to human and animal health

Conclude by highlighting the complexity of ARG sources and the need for interdisciplinary collaborations between environmental scientists, public health officials, wastewater management authorities, and agricultural stakeholders to address the issue of antibiotic resistance in urban river ecosystems effectively

Check name of microbes and write all of them in italics

Cross Check all references

Fig. 1 indicate standard error or deviation calculated.  

 Moderate editing of English language required

Author Response

In introduction section, Emphasize the significance of untreated sewage as a primary source of tetracycline resistance genes (tetO and tetW) in the urban river impacted by combined sewage overflow (CSO) events.            

We appreciate the suggestion and have included three new sources in lines 22-24 that specifically address CSOs as a source of ARGs in urban waterways.

Also Highlight the need for improved wastewater treatment to mitigate the introduction of antibiotic resistance genes (ARGs) into the aquatic environment

We added language to address wastewater infrastructure in lines 94-101 and 202-209.

Discuss the potential health implications of high abundances of tetracycline resistance genes in the CSO-impacted site. Explore the possibility of gene transfer and the spread of antibiotic resistance among bacteria in the environment, posing risks to human and animal health

We added a source to the first reference of HGT in the manuscript (line 17) and language discussing gene transfer and risk to human health in the conclusion section (Lines 197-209).

Conclude by highlighting the complexity of ARG sources and the need for interdisciplinary collaborations between environmental scientists, public health officials, wastewater management authorities, and agricultural stakeholders to address the issue of antibiotic resistance in urban river ecosystems effectively

Based on your suggestion, we have added these points to the conclusions (lines 197-209), and appreciate the opportunity it creates to synthesize our results in a broader context.

Check name of microbes and write all of them in italics

Done

Cross Check all references

All references have been checked and formatted to match the journal guidelines.

Fig. 1 indicate standard error or deviation calculated.

There is no standard error or deviation associated with this figure. The graph is a frequency histogram, which reports how many observations (as a % of total observations) fall within each abundance range/bin.  We added clarifying language in the Figure 1 legend (lines 66-67) and in the methods section (lines 330-334) to address this comment.

Reviewer 4 Report

Dear Editor and Authors

Faced with the great challenge related to environmental resistance to antibiotics, I consider the manuscript brings interesting results. However, some limitations are the small number of primers used, the lack of sequencing results and the time gap from the present related to sampling. In this sense, the study would gain more importance if it made a comparison between the 2015-2016 sampling and the current moment. Considering these limitations, I consider the manuscript should be published in another MDPI journal with a lower impact factor.

Minor comments

Affiliation: Please, include the country

Figure 2: It is necessary to include North indication in both images and include kilometer scale in the left image.

Author Response

Faced with the great challenge related to environmental resistance to antibiotics, I consider the manuscript brings interesting results. However, some limitations are the small number of primers used, the lack of sequencing results and the time gap from the present related to sampling. In this sense, the study would gain more importance if it made a comparison between the 2015-2016 sampling and the current moment. Considering these limitations, I consider the manuscript should be published in another MDPI journal with a lower impact factor.

We appreciate the reviewer's concern about the time gap, and agree that a study comparing 2015/16 to present conditions could be very interesting. However, that would address a different question – focusing on how ARG abundance changes over broad time scales. Our question instead focuses on sources, and we present a successful approach for differentiating multiple ARG sources within a relatively small area. The methodology we propose and overarching conclusions are still highly relevant despite the publication delay.

Affiliation: Please, include the country

Country information is now included in the affiliation.

Figure 2: It is necessary to include North indication in both images and include kilometer scale in the left image. 

Thank you for this suggestion. North arrows and scales are now present in both maps of Figure 2.

Reviewer 5 Report

The article "Drivers of Antibiotic Resistance Gene Abundance in an Urban River" shows interesting results, however I consider that the conclusion should be improved. It should be in accordance with the results obtained, highlighting its contribution.

Author Response

The article "Drivers of Antibiotic Resistance Gene Abundance in an Urban River" shows interesting results, however I consider that the conclusion should be improved. It should be in accordance with the results obtained, highlighting its contribution.

To address this concern, we synthesized our results and included more text in the discussion section (Lines 197-209), as well as rewriting the last sentence of the abstract.

Round 2

Reviewer 1 Report

Line 289-90 and 320-321 does not contain limit of detection (LOD). Please read Bustin et al., where it has guidelines for minimum things to report when using qpcr (https://doi.org/10.1373/clinchem.2008.112797).  Since there are many non-detects it is very important to report LOD values. The author have added the lowest concentration used to make standard curves but that is not the LOD value. I had previously asked if the concentration is in fact 0 or below the limit of detection? The x-axis in figure 1 says 0. The authors do not consider it could be  lower than the LOD value? Do the authors think that non-detects are really 0 as shown in Figure 1? Please report LOD value in terms of copies/ml as in figure 1. 

Author Response

Based on the reviewer’s concern, we have updated Figure 1 so that the X axis no longer says zero. We have replaced this category with  “ND” (not detected), and added language to the caption of Figure 1 that indicates this category represents samples that did not produce amplicons. We appreciate the reviewer’s persistence in making sure the information in Figure 1 is property contextualized, and for helping us avoid misleading the reader by reporting data as “0.”

In regards to the reviewer’s request that we review MIQE guidelines, we have included the vast majority of the applicable recommended items and are confident that we have provided a thorough documentation of our methods to the point any researcher could appropriately replicate our study. We utilized replications for our 8 point standard curves for each gene, multiple negative test controls, and melt curves in addition to gel electrophoresis to assure correct amplicon size and identity. We also provide complete reaction conditions, reaction volume, all concentrations and manufacturers, PCR efficiency, and r2.

With specific regards to LOD, Bustin et al. (2009) do not give a clear guidelines on how to calculate LOD. In fact, they state that “Appropriate determination and modeling of the LOD in the qPCR is the focus of continued research.” We have consulted several more recent papers and, unfortunately, find none of their approaches to be suitable for our dataset.

For example, the conventional approach is to rely on the background signal from blanks (detailed in Forootan et al. 2017, doi: 10.1016/j.bdq.2017.04.001). But, in our assays, no Cq values were obtained for any of the blanks, so it is not possible for us to estimate LOD using our existing data.  Further, this approach assumes the response is linear and data are normally distributed; qPCR assays fail to meet these assumptions. An alternative that has been recently published requires running a large number of replicate standard curves (Klymus et al. 2020,  doi.org/10.1002/edn3.29). Unfortunately, this is no longer an option due to the time elapsed since the study was conducted, but we will consider it for future work.

Like the reviewer, we appreciate that is important for the reader to have some context for how many target copies each assay can detect, so we elected to add information on lowest point on the standard curves during the previous revision. We used the units reported in the Klymus et al. 2020 (doi.org/10.1002/edn3.29) are careful not to mislead the reader by calling this LOD. We feel including these values is a substantial benefit compared to the large number of studies that do not include LOD or any quantitative framework for gene copy results (e.g., doi.org/10.3390/w13192733, doi.org/10.1016/j.scitotenv.2017.10.128,doi.org/10.1021/acs.est.7b03380). We believe this information, combined with the changes made to Figure 1, have sufficiently addressed the reviewer’s concerns.

Reviewer 4 Report

no comments

Author Response

Thank you